# The Protective Effects of *Acer okamotoanum* and Isoquercitrin on Obesity and Amyloidosis in a Mouse Model

**DOI:** 10.3390/nu12051353

**Published:** 2020-05-09

**Authors:** Ji Hyun Kim, Sanghyun Lee, Eun Ju Cho

**Affiliations:** 1Department of Food Science and Nutrition & Kimchi Research Institute, Pusan National University, Busan 46241, Korea; kjjjjhh11@naver.com; 2Department of Plant Science and Technology, Chung-Ang University, Anseong 17546, Korea; slee@cau.ac.kr

**Keywords:** *Acer okamotoanum*, amyloid beta, high fat diet, isoquercitrin, obesity

## Abstract

Obesity increases risk of Alzheimer’s Disease (AD). A high fat diet (HFD) can lead to amyloidosis and amyloid beta (Aβ) accumulation, which are hallmarks of AD. In this study, protective effects of the ethyl acetate fraction of *Acer okamotoanum* (EAO) and isoquercitrin were evaluated on obesity and amyloidosis in the HFD- and Aβ-induced mouse model. To induce obesity and AD by HFD and Aβ, mice were provided with HFD for 10 weeks and were intracerebroventricularly injected with Aβ_25–35_. For four weeks, 100 and 10 mg/kg/day of EAO and isoquercitrin, respectively, were administered orally. Administration of EAO and isoquercitrin significantly decreased body weight in HFD and Aβ-injected mice. Additionally, EAO- and isoquercitrin-administered groups attenuated abnormal adipokines release via a decrease in leptin and an increase in adiponectin levels compared with the control group. Furthermore, HFD and Aβ-injected mice had damaged liver tissues, but EAO- and isoquercitrin-administered groups attenuated liver damage. Moreover, administration of EAO and isoquercitrin groups down-regulated amyloidosis-related proteins in the brain such as β-secretase, presenilin (PS)-1 and PS-2 compared with HFD and Aβ-injected mice. This study indicated that EAO and isoquercitrin attenuated HFD and Aβ-induced obesity and amyloidosis, suggesting that they could be effective in preventing and treating both obesity and AD.

## 1. Introduction

For many years now, obesity has been a worldwide public health problem. The imbalance of energy such as excessive intake of a high fat diet (HFD) is a major factor for obesity [1,2]. It leads to increased fat mass, changes in lipid profiles, and abnormalities in adipokine release in the body [3]. In particular, obesity increases the risk of many diseases such as diabetes mellitus, cardiovascular disease, and Alzheimer’s disease (AD) [2,4]. Recent studies demonstrated a correlation between obesity and AD. HFD has been shown to strongly increase the risk of AD development [5]. In addition, obesity induced by HFD exacerbated cognitive and memory decline, whereas a low fat diet led to cognitive and memory improvement effects in mouse models in vivo [6]. Furthermore, HFD-induced mice indicated AD-related pathology such as accumulation of Aβ and presence of neurofibrillary tangles (NFT) in the brain tissue [7,8]. In particular, consumption of HFD increases amyloidosis-related factors such as beta-site amyloid beta precursor protein (APP) cleaving enzyme-1 (BACE-1), which elevates accumulation of Aβ plaques [8,9]. HFD also induces oxidative stress, inflammation, brain insulin resistance, and synaptic dysfunction in cognitive function, and exacerbates cognitive dysfunction [9,10,11]. Obesity in AD mice aggravates cognitive impairment by metabolic changes and Aβ accumulation, compared with AD mice who eat a normal diet (ND) [12,13]. Previous studies have reported that developing therapeutic agents can help in preventing both obesity and AD using HFD and Aβ-induced mouse model [14,15,16].

*Acer okamotoanum*, a natural endemic species in the Republic of Korea, has several important biological properties such as antimicrobial properties, antioxidant properties, and neuroprotective properties [17,18,19,20]. A leaf of *A. okamotoanum* showed anti-obesity effects by suppressing adipogenesis under the cellular system [21]. The ethyl acetate fraction of *A. okamotoanum* (EAO) contains several bioactive flavonoids including quercitrin, isoquercitrin, and afzelin [22]. In addition, we previously confirmed that isoquercitrin isolated from EAO had higher anti-obesity effects via inhibition of adipogenesis and induction of lipolysis in 3T3-L1 cells, among other flavonoids from EAO. Furthermore, administration of EAO and isoquercitrin improved cognitive function via attenuation of oxidative stress in HFD and Aβ-induced mice in vivo [23]. However, the effect of EAO and its active compounds on cognitive dysfunction induced by both obesity and amyloidosis in in vivo mouse models has not been investigated. Donepezil and conjugated linoleic acid (CLA) are widely used in treatment of AD and obesity, respectively [24,25]. Therefore, this study focused on the protective effects of EAO and isoquercitrin on obesity and amyloidosis compared with donepezil and CLA in an HFD- and Aβ-induced mouse model in vivo. 

## 2. Materials and Methods 

### 2.1. Reagents

Aβ_25–35_ was purchased from Sigma Aldrich (St. Louis, MO, USA), dissolved in saline solution and incubated at 37 °C for 72 h to aggregate Aβ_25–35_. Aggregated Aβ_25–35_ solution was administered via intracerebroventricular (i.c.v.) injection to mice. Donepezil (#D6821) was purchased from Sigma Aldrich (St. Louis, MO, USA). CLA was purchased from Novarex Co. (Cheongju, Korea). 

### 2.2. Sample Preparation

EAO and isoquercitrin were prepared according to Lee et al. [22]. In brief, aerial parts of *A. okamotoanum* were dried and subjected to extraction using methanol (MeOH) under a reflux system. MeOH extract was suspended in distilled water and partitioned to obtain EAO. Isoquercitrin was isolated from this EAO by repeated column chromatography.

### 2.3. Animals

Five-week-old male C57BL/6J mice (20 ± 2 g) were purchased from Central Laboratory Animal Inc. (Seoul, Korea). The mice were conditioned in a maintained temperature (20 °C ± 2 °C) and humidity (50% ± 10%) under a 12 h light/dark cycle. The animal protocol used in this study was reviewed and approved (No. PNU-2017-1481) by the Pusan National University Institutional Animal Care and Use Committee (PNU-IACUC). During the entire experimental period, mice were provided with ad libitum access to water and diet.

### 2.4. Groups and Treatment

Mice were fed with a commercially available ND (#AIN76-A, Research Diets, New Brunswick, NJ, USA) during the one week adaptation period. Obesity was induced by feeding the mice with HFD (60% kcal from fat, #D12492, Research Diets, New Brunswick, NJ, USA) for 10 weeks. After consumption of the HFD for 10 weeks, mice were divided into six groups (*n* = 6) and were administered with saline solution or Aβ_25–35_ via i.c.v. injection. After three days adaptation, four groups with HFD- and Aβ_25–35_-induced mice were orally administered with EAO (100 mg/kg/day), isoquercitrin (10 mg/kg/day), donepezil (5 mg/kg/day), and CLA (750 mg/kg/day) dissolved in 0.5% carboxymethylcellulose (CMC) for four weeks. The experimental groups were divided into six groups as follows: (1) normal, mice fed with ND, injected saline solution, and orally administered 0.5% CMC; (2) control, mice fed with HFD, injected Aβ_25–35_, and orally administered 0.5% CMC; (3) AO, mice fed with HFD, injected Aβ_25–35_, and orally administered EAO; (4) IQ, mice fed with HFD, injected Aβ_25–35_, and orally administered isoquercitrin; (5) DO, mice fed with HFD, injected Aβ_25–35_, and orally administered donepezil; (6) CLA, mice fed with HFD, injected Aβ_25–35_, and orally administered CLA. Food intakes and body weight were consistently measured every day during the experiment. The food efficiency ratio (FER) was calculated as body weight gain per gram of food consumed. The mice were overexposed to CO_2_ gas to be anaesthetized, and blood was collected by cardiac puncture. Mouse tissue such as liver, brain, and adipose tissue were perfused with saline solution and removed. The schedule of the entire experiment is shown in Figure 1.

### 2.5. Aβ_25–35_ i.c.v. Injection

According to the procedure established by Laursen and Belknap, mice were administered with Aβ_25–35_ via i.c.v. [26]. Briefly, the mice were anesthetized with Zoletil^50^ and Rumpun mixture (3:1 ratio) intraperitoneally and placed in a stereotaxic apparatus. Mice were i.c.v. injected with aggregated Aβ_25–35_ (25 nM/5 µL) or saline solution from bregma into the lateral ventricle at the following coordinates: anterior/posterior −2.0 mm, medial/lateral 1.5 mm, and dorsal/ventral 2.0 mm, according to the stereotaxic atlas of mouse brain, using a microinfusion pump (Baoding Longer Precision Pump Co., Ltd., Baoding, China) [27].

### 2.6. Biochemical Analysis

The blood samples were incubated for 30 min at room temperature and centrifuged at 1164× *g* at 4 °C for 15 min. Serum leptin and adiponectin were measured using mouse leptin ELISA kit (ab100718, Abcam, Cambridge, United Kingdom) and mouse/rat high molecular weight adiponectin ELISA kit (AKMAN-011, Shibayagi, Gunma, Japan), according to the manufacturer’s protocols. Blood glucose was measured by automatic chemical analyzer (Hitachi 7600, Hitachi Co., Tokyo, Japan). 

### 2.7. Histopathological Analysis

The liver and adipose tissues were removed from each mouse and fixed in 10% (*v/v*) formalin solution. Fixed samples were embedded in paraffin in preparation for staining with hematoxylin and eosin (H&E). Stained slices were viewed under an optical microscope (ECLIPSETi; Nikon Corp., Tokyo, Japan) with magnification 100 times. 

### 2.8. Western Blotting 

Whole brain tissue was homogenized and lysed in ice cold RIPA lysis buffer, containing protease inhibitor cocktail for 30 min at 4 °C. The lysates were then centrifuged at 18,627× *g* for 30 min at 4 °C, and supernatants were separated. The protein quantification was measured using Bio-Rad Assay kit (Bio-Rad, Hercules, CA, USA). An equal amount of protein samples (15 μg) were separated by 10% or 13% sodium dodecyl sulphate-polyacrylamide gel electrophoresis and transferred to polyvinylidene difluoride membranes (Merck Millipore Ltd., Billerica, MA, USA). The membranes were blocked using 5% skim milk solution for 1 h at room temperature and incubated with primary antibodies such as BACE-1, presenilin (PS)-1, PS-2, and β-actin overnight at 4 °C. The membranes were then washed with phosphate buffered saline-tween (PBS-T) and incubated with horseradish peroxidase (HRP)-conjugated secondary antibody for 1 h at room temperature. Visualization of the bands was done by using an enhanced chemiluminescence detection solution and a Davinch-chemi^TM^ Chemiluminescence Imaging System (Davinch-K, Seoul, Korea). The protein band intensities were quantified using Image J software. In addition, the density of bands was expressed as a ratio of the band density divided by that of the hose-keeping protein, β-actin.

### 2.9. Statistical Analysis

All data were presented as means ± SD (standard deviation) and were analyzed using one-way ANOVA followed by Duncan’s multiple test using SPSS program (SPSS Inc. Chicago, IL, USA, version 20). *p* < 0.05 was considered to indicate a statistically significant difference among all groups.

## 3. Results

### 3.1. Effect of EAO and Isoquercitrin on Body Weight

The body weight change, body weight gain, histopathological change in adipose tissue, and FER of the mice are shown in Figure 2. The body weights were monitored in all groups during the experimental period. Initial body weights showed statistically insignificant differences among experimental groups. During the experimental period, the ND and saline solution-induced normal group showed 6.55 g of body weight gain, whereas the HFD-fed and Aβ-injected control group showed 26.75 g of body weight gain. However, administration of EAO, isoquercitrin, donepezil, and CLA decreased body weight gain during four weeks compared with the control group. EAO- and isoquercitrin-administered groups showed 22.78 g and 19.77 g of body weight gain, respectively. Furthermore, the CLA-administered group indicated 21.90 g of body weight gain. In addition, histological results revealed that the size of adipocytes in adipose tissue was significantly increased in the HFD and Aβ-induced control group, compared with the ND and saline solution-induced normal group. However, the size of adipocytes in adipose tissue decreased significantly upon administration of EAO and isoquercitrin compared with the control group. The group administered with CLA also observed a reduction in the size of adipocytes. Furthermore, FER levels in the control group were high, whereas groups that were administered EAO and isoquercitrin had significantly reduced FER levels. The CLA-administered mice showed a significant decrease in FER levels compared with the control group.

### 3.2. Effect of EAO and Isoquercitrin on Blood Glucose Level

As shown in Figure 3, the blood glucose levels of the control group were significantly higher (238.17 mg/dL) than the normal group (132.83 mg/dL). However, administration of EAO and isoquercitrin significantly inhibited blood glucose levels to 185.33 mg/dL and 171.00 mg/dL compared with the control group. In addition, administration of CLA significantly reduced blood glucose levels to 168.50 mg/dL compared with the control group.

### 3.3. Effect of EAO and Isoquercitrin on Adipokines Changes

Figure 4 showed the effects of EAO and isoquercitrin on adipokines changes in the HFD- and Aβ-induced mouse models. The serum leptin level of the normal group observed 15.47 ng/mL, whereas the control group significantly increased leptin level to 158.23 ng/mL. However, administration of EAO and isoquercitrin significantly decreased serum leptin levels to 63.84 ng/mL and 90.43 ng/mL, respectively, compared with the control group. In addition, CLA-administered mice significantly inhibited serum leptin levels of 78.49 ng/mL. The adiponectin level of the normal group indicated 193.83 ng/mL, but the control group induced by HFD and Aβ indicated 91.13 ng/mL. However, AO and IQ groups showed significantly higher adiponectin levels, 183.04 ng/mL and 167.27 ng/mL, respectively. Furthermore, CLA groups also increased adiponectin levels of 178.60 ng/mL. The administration of EAO and isoquercitrin showed almost similar effects on adipokine of CLA, a positive control against obesity.

### 3.4. Effect of EAO and Isoquercitrin on Hepatic Damage

We investigated the effects of EAO and isoquercitrin on hepatic damage in the HFD and Aβ- induced mice. The serum AST and ALT levels and histological changes of liver tissue in all experimental groups were observed (Figure 5). The normal group serum was found to have an AST level of 99.48 karmen, but the control group was found to have 136.04 karmen. However, the level of AST in serum of isoquercitrin-administered group (95.57 karmen) was significantly lower compared with the control group. On the other hand, AO, DO, and CLA groups did not observe a statistically significant reduction in AST levels compared with the control group. The ALT level in the serum of the normal group was 17.54 karmen, while the control group increased to 94.87 karmen. However, AO (58.46 karmen) and the IQ group (39.84 karmen) significantly decreased their ALT level compared with the control group. The donepezil- and CLA-administered group did not show a reduction in ALT levels compared with the control group. Moreover, histopathological examinations of the liver showed an increase in steatosis grade in the control groups, while administration of EAO and isoquercitrin groups showed a decrease in steatosis grade.

### 3.5. Effect of EAO and Isoquercitrin on β-Secretase Activity in the Brain Tissue

To investigate the influence of β-secretase activity on the amyloidosis of EAO and isoquercitrin, we measured BACE-1 protein expression in the brain tissue. As shown in Figure 6, the Aβ-injected and HFD-induced control group showed significant elevation of BACE-1 compared with the normal group. On the other hand, administration of EAO and isoquercitrin in Aβ-injected and HFD-induced mice showed decreased protein expression of BACE-1 compared with the control group. In addition, mice administered with donepezil showed a significant decrease in BACE-1 expression. 

### 3.6. Effect of EAO and Isoquercitrin on γ-Secretase Activity in the Brain Tissue

Figure 7 indicates the effect of EAO and isoquercitrin on γ-secretase activity of amyloidosis in the brain tissue, where we measured protein expression of PS-1 and PS-2 components of γ-secretase. The control group induced by HFD and Aβ significantly elevated protein expression of both PS-1 and PS-2, compared with the normal group. However, as shown in Figure 7A, the EAO-administered group significantly inhibited protein expression of PS-1 compared with the control group. Additionally, the EAO-administered group inhibited higher protein expression of PS-1 than the donepezil-administered group which was used as a positive control against AD. However, the isoquercitrin-administered group did not observe statistical reduction in PS-1 expression compared with the control group. As shown in Figure 7B, the isoquercitrin-administered group significantly inhibited protein expression of PS-2 compared with the control group, but the EAO-administered group did not have a significant reduction in PS-2. Furthermore, administration of isoquercitrin was inhibited by PS-2 protein expression in the brain, similar to the administration of donepezil.

## 4. Discussion

The prevalence of obesity and the risk of AD has increased worldwide [28]. Recently, many studies have reported a correlation between obesity and AD. It has been reported that obesity induced by HFD leads to cognitive impairment in human and animal models [28,29]. AD is the most common type of dementia that indicates cognitive impairment, and the accumulation of Aβ in the brain is one of its pathological hallmarks [7,8]. Amyloidosis, which produces Aβ cleaved from APP by activation of β- and γ-secretase, is known as the representative cause of AD [7,30]. In particular, several studies indicate that consumption of HFD accelerates accumulation of Aβ, thereby promoting cognitive impairment by increasing neuronal oxidative stress and neuroinflammation in an AD mouse model [31,32]. HFD-induced obesity in the AD mouse model has indicated cognitive impairment via elevation of Aβ accumulation in the brain, as well as abnormal metabolic conditions such as increased body weight and deterioration of glucose tolerance and liver damage, compared with AD mice who consumed a normal diet [12,33]. Therefore, HFD-induced obesity in AD mouse models is widely used to study prevention and treatment of both obesity and AD. In this study, we evaluated the effect of EAO and isoquercitrin on cognitive impairment using an HFD and Aβ_25–35_-induced mouse model.

EAO contains various flavonoids which include quercitrin, isoquercitrin, and afzelin [22]. Previous studies reported that EAO protected against Aβ_25–35_-induced glial cell damage via inhibition of reactive oxygen species production [34]. In addition, EAO improved cognition and memory function through attenuation of oxidative stress in Aβ_25–35_-induced in vivo AD mouse models [35]. Moreover, EAO and its three flavonoids exerted neuroprotective effects against oxidative stress-induced SH-SY5Y human neuronal cells [19,20]. In particular, isoquercitrin showed higher protective activity from neuronal cells against hydrogen peroxide-induced oxidative stress than other flavonoids in EAO such as quercitrin and afzelin [20]. In addition, leaves from *A. okamotoanum* inhibited adipogenesis in 3T3-L1 adipocytes [21]. We previously investigated the cognitive improvement effects of EAO and isoquercitrin on HFD and Aβ-induced cognitive impairment in mice [23]. However, the protective effects of EAO and isoquercitrin under HFD induced AD in vivo have not been investigated. Therefore, this study investigated the protective effects of EAO and isoquercitrin against obesity and amyloidosis under an HFD and Aβ_25–35_-induced in vivo mouse model. Donepezil is a cholinesterase inhibitor and it has been widely used to alleviate cognitive deficits in AD patients [24]. CLA is commonly known as a weight management agent for anti-obesity [25]. Donepezil and CLA were used as positive controls for anti-AD and anti-obesity, respectively. 

In this study, an obese mouse model was used by administrating HFD for a period of 10 weeks, and then i.c.v. injecting Aβ_25–35_ in obese mice. EAO, isoquercitrin, donepezil, and CLA were then orally administered to mice for four weeks. EAO and isoquercitrin reduced body weight gain, fat size in adipose tissue, and FER compared with the control group. In particular, EAO and isoquercitrin decreased body weight gain and FER, similar to the CLA-administered group. Several studies support the protective role of EAO and isoquercitrin from obesity under cellular and in vivo systems [22,36,37]. Kim et al. demonstrated that EAO showed anti-adipogenesis via down-regulation of PPARγ protein expression in the 3T3-L1 cell [22]. Isoquercitrin also attenuated adipogenesis via the inhibition of Wnt/β-catenin signaling in 3T3-L1 cells [36]. In addition, supplementation of isoquercitrin demonstrated inhibition of body weights via activation of AMP-activated protein kinase (AMPK) signaling in an HFD-induced mouse model [37].

Obesity induced by HFD is widely known to increase blood glucose and change adipokines levels [38]. In a previous study, HFD-induced AD mice confirmed elevation of blood glucose levels compared with an ND-induced AD mouse model [6]. Leptin and adiponectin are constituent of adipocytokines and are produced from adipose tissue; therefore, regulation of these factors is important in the development of metabolic diseases including obesity [38,39]. Leptin plays an important role in regulation of satiety and energy homeostasis, while adiponectin is an adipocyte-derived hormone that acts on anti-obesity, antidiabetic, and anti-inflammatory activity [40,41]. An increase in leptin and inhibition of adiponectin levels are both associated with obesity [39]. In addition, HFD-induced AD mice revealed hyperleptinemia and increased leptin levels, compared with ND-induced AD mice [42,43]. In this study, administration of EAO and isoquercitrin decreased blood glucose levels compared with the control group. In particular, IQ groups showed reduced blood glucose levels similar to the CLA group. In addition, EAO and isoquercitrin-administered mice decreased leptin levels, but increased adiponectin levels were found in HFD and Aβ-induced mice compared with the non-administrated control group. In particular, EAO and isoquercitrin-administered mice attenuated abnormal adipokines release, similar to CLA groups, via regulation of leptin and adiponectin release. Therefore, we suggest that EAO and isoquercitrin attenuated abnormal changes of adipokines such as leptin and adiponectin in HFD and Aβ-induced mice. In particular, isoquercitrin elevated protein expression of adiponectin receptors such as AdipoR1 and AdipoR2 in the rat hepatoma cells [44]. Therefore, these results show that EAO and isoquercitrin promote anti-obesity activity by regulating lipid and glucose metabolism in HFD- and Aβ-induced mice.

HFD induces liver damage by regulation of specific markers of liver function such as ALT and AST in mice [3,43]. HFD-induced AD mice also had increased AST and ALT levels, as well as hepatic steatosis [43]. In our results, administration of EAO significantly decreased ALT levels. Administration of isoquercitrin significantly inhibited both AST and ALT levels compared with HFD- and Aβ-induced control mice. However, administration of donepezil and CLA did not statistically decrease AST and ALT levels. To investigate fatty accumulation and liver toxicity of samples, we investigated the liver pathology by H&E staining. These results showed approximate fatty liver lesions and toxicity. As shown in Figure 5C, the HFD-fed control group showed an accumulation of numerous fatty droplets and hepatic steatosis compared with the ND-fed normal group. The administration of sample groups attenuated accumulation of liver fatty droplets and hepatic steatosis. Further study is needed on the hepatoprotective effects and the molecular mechanism of samples by quantify ballooning, fibrosis, and accumulation of lipids or by measuring inflammatory markers. A previous study demonstrated that administration of isoquercitrin for five weeks inhibited hepatic inflammation and oxidative stress by suppressing transforming growth factor beta (TGF-β), signaling in the HFD-induced mice [37]. In addition, the sap of EAO reduced the hepatic oxidative stress and apoptosis in the alcohol-induced hepatic damage mouse model [45]. Therefore, our results suggest that EAO and isoquercitrin attenuate HFD-induced liver toxicity in HFD- and Aβ-induced mice.

Nonalcoholic fatty liver disease (NAFLD) is characterized by accumulation of fat in liver, and it is known to be associated with AD. NAFLD mice induced by HFD showed cognitive and memory impairment in behavior tests [46]. The HFD-fed NAFLD mice showed decreased Aβ clearance in the liver, followed by induced aggravation of cognitive impairment by accumulation of Aβ plaques in the brain [46,47]. The HFD-fed mice significantly increased circulation of Aβ levels by decreasing low-density lipoprotein receptor-1 [33,47]. In addition, fatty liver increased neuroinflammation by increasing microgliosis, toll-like receptors, and inflammatory cytokines in the brain [33]. Furthermore, fatty liver-induced mice increased neuronal oxidative stress by increasing gene expression of NADPH oxidase 2 [46]. Therefore, fatty liver is closely related to AD progression by regulation of Aβ clearance, neuroinflammation, and oxidative stress. 

Amyloidosis is a process of Aβ production cleaved from APP [48]. Aβ cleaved from APP by β- and γ-secretase accumulates in the patient’s brain and produces Aβ plaque, which is known to cause AD. HFD is known to elevate Aβ accumulation in the brain by induction of oxidative stress via over-production of reactive oxygen species [49]. Several studies indicate that consumption of HFD in AD mice elevates Aβ accumulation, which induces oxidative stress via down-regulation of nuclear factor erythroid 2-related factor 2 (Nrf-2) signaling, inhibition of antioxidant enzymes such as superoxide dismutase, and induction of lipid peroxidation in the brain [49,50]. Furthermore, elevated Aβ levels by consumption of HFD leads to inflammation by disrupting the blood–brain barrier and increasing inflammatory cytokines, resulting in induction of neuronal apoptosis via up-regulation of B-cell lymphoma 2-associated X (Bax) protein in the brain tissue [14,50,51]. Therefore, reduction of amyloidosis is an important role for prevention and treatment of AD in the HFD- and Aβ-induced mouse model. Recently, natural products have been attracting much attention for treatment and prevention of AD via regulation of amyloidosis process [52]. In the amyloidosis process, APP is cleaved by BACE-1 (member of β-secretase) and generates sAPPβ and CTF-β [48]. BACE-1 expression promotes the production of the β-secretase enzyme which leads to increased Aβ production [53]. Obesity induced by HFD-fed rats for 16 weeks significantly increased BACE-1 gene expression in the brain [8]. In addition, consumption of HFD elevated Aβ oligomer and deposited Aβ levels via up-regulation of CTF-β in the brain of AD-induced mice compared with ND-consuming AD mice [12]. In our study, both EAO and isoquercitrin inhibited protein expression of BACE-1, thereby EAO and isoquercitrin attenuated amyloidosis induced by HFD and Aβ in the brain by down-regulation of β-secretase activity. 

CTF-β cleaved by β-secretase from APP is metabolized by γ-secretase and then produces Aβ [48]. γ-Secretase consists of four transmembrane proteins, PS (consists of PS-1 and PS-2), presenilin enhancer-2, nicastrin, and anterior pharynx-defective 1 [54]. Mutation of PS leads to increased accumulation of Aβ which is also considered a major cause of AD [55]. Several studies have shown that HFD-fed mice exhibit significantly higher expression of PS-1 and PS-2 in the brain compared with ND-fed mice [56]. With respect to γ-secretase activity in our study, PS-1 is down-regulated by EAO, and PS-2 is inhibited by isoquercitrin. In particular, EAO showed higher inhibition of PS-1 expression and isoquercitrin had a similarly reduced PS-2 expression compared with the positive control group by administration of donepezil. Therefore, EAO and isoquercitrin suppressed amyloidosis in the brain by down-regulation of γ-secretase activity as a result of PS-1 and PS-2 inhibition, respectively. 

## 5. Conclusions

Our results showed that EAO and its active compounds—mainly isoquercitrin—attenuated obesity by reducing body weight, changing adipokines, and inhibiting liver damage in HFD- and Aβ-induced mouse models. In addition, EAO and isoquercitrin inhibited amyloidosis in the brain by down-regulating amyloidosis-related protein. This study suggests that EAO and isoquercitrin could be an agent of treatment and prevention for both obesity and AD. 

## Figures and Tables

**Figure 1 nutrients-12-01353-f001:**
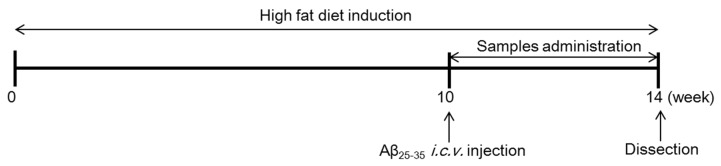
Experimental schedule.

**Figure 2 nutrients-12-01353-f002:**
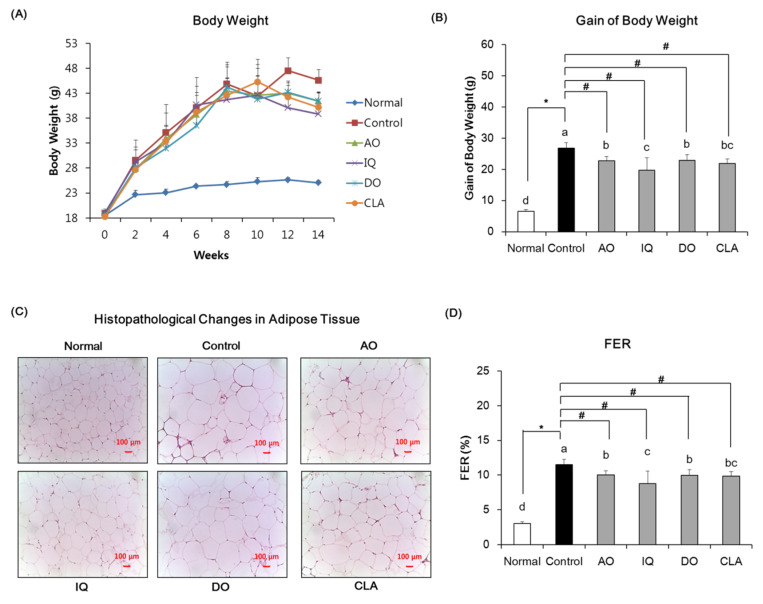
Changes in body weight (**A**), gain of body weight (**B**), histopathological changes in adipose tissue (**C**), and food efficiency ratio (**D**). FER: food efficiency ratio (body weight gains per gram of food consumed). Normal: normal diet + saline solution + 0.5% CMC; Control: high fat diet (HFD) + Aβ_25–35_ + 0.5% CMC; AO: HFD + Aβ_25–35_ + EAO (100 mg/kg/day); IQ: HFD + Aβ_25–35_ + isoquercitrin (10 mg/kg/day); DO: HFD + Aβ_25–35_ + donepezil (5 mg/kg/day); CLA: HFD + Aβ_25–35_ + CLA (750 mg/kg/day). Scale bar, 100 μM. Values are mean ± SD (*n* = 6). ^a–d^ Means indicated with different letters are significantly different by Duncan’s multiple range test among all groups. * *p* < 0.05 vs. normal group; ^#^
*p* < 0.05 vs. control group.

**Figure 3 nutrients-12-01353-f003:**
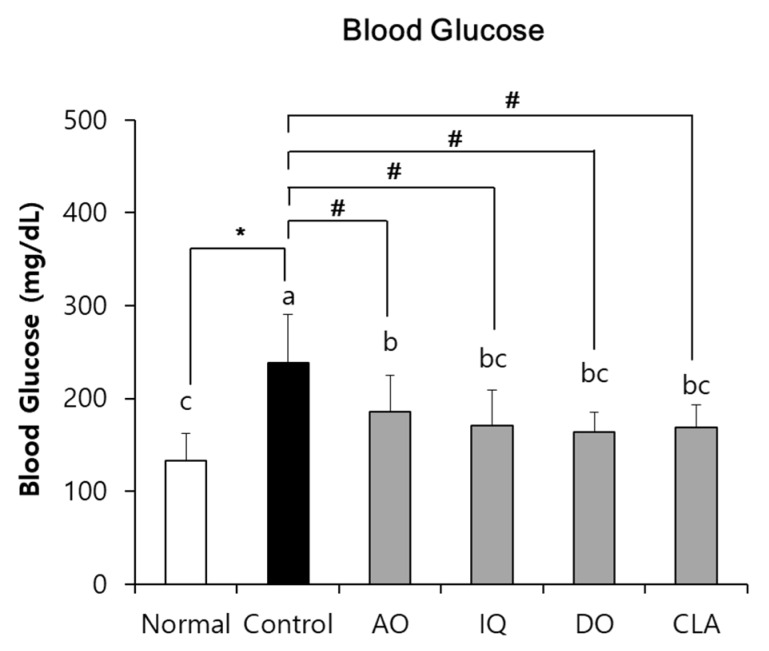
The effect of EAO and isoquercitrin on blood glucose level. Normal: normal diet + saline solution + 0.5% CMC; Control: high fat diet (HFD) + Aβ_25–35_ + 0.5% CMC; AO: HFD + Aβ_25–35_ + EAO (100 mg/kg/day); IQ: HFD + Aβ_25–35_ + isoquercitrin (10 mg/kg/day); DO: HFD + Aβ_25–35_ + donepezil (5 mg/kg/day); CLA: HFD + Aβ_25–35_ + CLA (750 mg/kg/day). Values are mean ± SD (*n* = 6). ^a–c^ Means indicated with different letters are significantly different by Duncan’s multiple range test among all groups. * *p* < 0.05 vs. normal group; ^#^
*p* < 0.05 vs. control group.

**Figure 4 nutrients-12-01353-f004:**
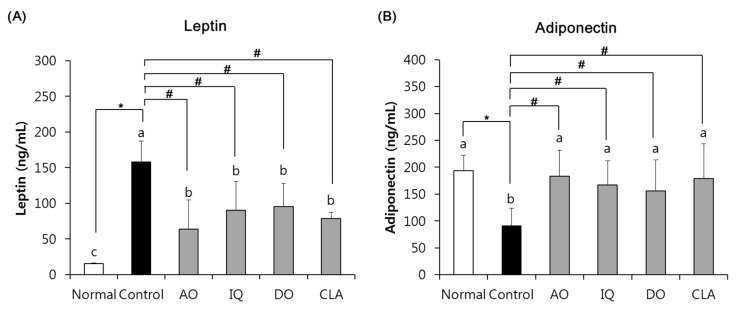
The effect of EAO and isoquercitrin on serum leptin (**A**) and adiponectin (**B**) levels. Normal: normal diet + saline solution + 0.5% CMC; Control: high fat diet (HFD) + Aβ_25–35_ + 0.5% CMC; AO: HFD + Aβ_25–35_ + EAO (100 mg/kg/day); IQ: HFD + Aβ_25–35_ + isoquercitrin (10 mg/kg/day); DO: HFD + Aβ_25–35_ + donepezil (5 mg/kg/day); CLA: HFD + Aβ_25–35_ + CLA (750 mg/kg/day). Values are mean ± SD (*n* = 6). ^a–c^ Means indicated with different letters are significantly different by Duncan’s multiple range test among all groups. * *p* < 0.05 vs. normal group; ^#^
*p* < 0.05 vs. control group.

**Figure 5 nutrients-12-01353-f005:**
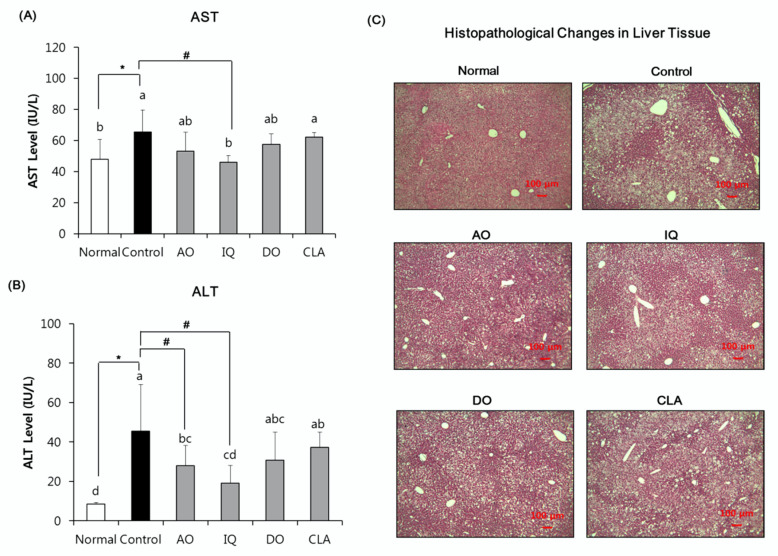
The Effect of EAO and isoquercitrin on the serum AST (**A**), serum ALT (**B**), and histological changes of liver tissue (**C**). Normal: normal diet + saline solution + 0.5% CMC; Control: high fat diet (HFD) + Aβ_25–35_ + 0.5% CMC; AO: HFD + Aβ_25–35_ + EAO (100 mg/kg/day); IQ: HFD + Aβ_25–35_ + isoquercitrin (10 mg/kg/day); DO: HFD + Aβ_25–35_ + donepezil (5 mg/kg/day); CLA: HFD + Aβ_25–35_ + CLA (750 mg/kg/day). Scale bar, 100 μM. Values are mean ± SD (*n* = 6). ^a–d^ Means indicated with different letters are significantly different by Duncan’s multiple range test among all groups. * *p* < 0.05 vs. normal group; ^#^
*p* < 0.05 vs. control group.

**Figure 6 nutrients-12-01353-f006:**
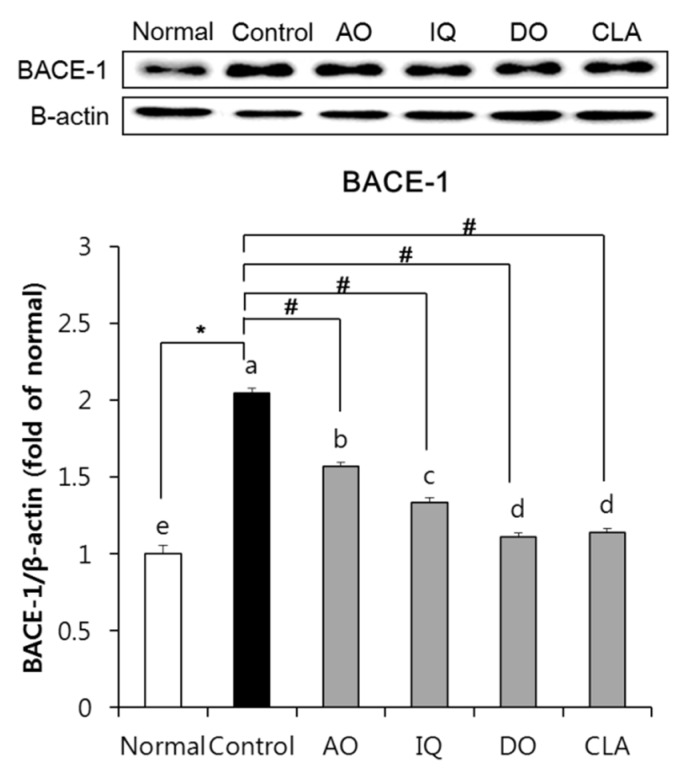
The effect of EAO and isoquercitrin on BACE-1 protein expression in the brain tissue. β-actin was used as loading control. Normal: normal diet + saline solution + 0.5% CMC; Control: high fat diet (HFD) + Aβ_25–35_ + 0.5% CMC; AO: HFD + Aβ_25–35_ + EAO (100 mg/kg/day); IQ: HFD + Aβ_25–35_ + isoquercitrin (10 mg/kg/day); DO: HFD + Aβ_25–35_ + donepezil (5 mg/kg/day); CLA: HFD + Aβ_25–35_ + CLA (750 mg/kg/day). Values are mean ± SD (*n* = 6). ^a–e^ Means indicated with different letters are significantly different by Duncan’s multiple range test among all groups. * *p* < 0.05 vs. normal group; ^#^
*p* < 0.05 vs. control group.

**Figure 7 nutrients-12-01353-f007:**
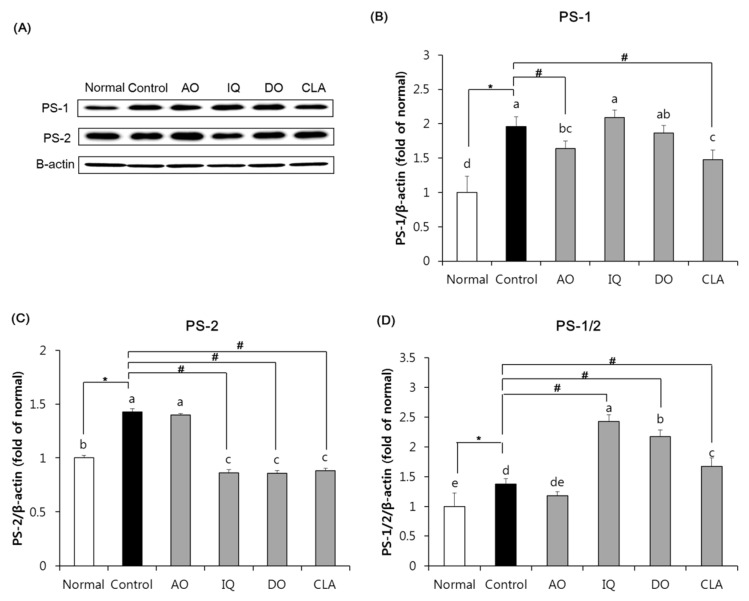
The effect of EAO and isoquercitrin on PS-1 and PS-2 protein expression in the brain tissue. Western blot band (**A**), PS-1 (**B**), PS-2 (**C**), and PS-1/2 ratio (**D**). The same membrane was used for expression of PS-1 and PS-2 with stripping and re-probing. β-actin was used as loading control. Normal: normal diet + saline solution + 0.5% CMC; Control: high fat diet (HFD) + Aβ_25–35_ + 0.5% CMC; AO: HFD + Aβ_25–35_ + EAO (100 mg/kg/day); IQ: HFD + Aβ_25–35_ + isoquercitrin (10 mg/kg/day); DO: HFD + Aβ_25–35_ + donepezil (5 mg/kg/day); CLA: HFD + Aβ_25–35_ + CLA (750 mg/kg/day). Values are mean ± SD (*n* = 6). ^a–e^ Means indicated with different letters are significantly different by Duncan’s multiple range test among all groups. * *p* < 0.05 vs. normal group; ^#^
*p* < 0.05 vs. control group.

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
