# Peer review of "The Protective Effects of Acer okamotoanum and Isoquercitrin on Obesity and Amyloidosis in a Mouse Model"

_nutrients, 2020, doi:10.3390/nu12051353_

Round 1

Reviewer 1 Report

The authors Kim et al. demonstrated the effect of A. okamotoanum and its active compound Isoquercitrin (IQ) along with positive controls for anti-AD and anti-obesity. This article is part of their already published article Kim et al. 2019.

  1. The rationale of using positive control mentioned in the discussion section. Please bring the write-up in the introduction section for a better understanding of readers. The write-up will provide information to readers why the donepezil and CLA used in the study.
  2. The primary concern of the article is the number of mice/samples used in each parameter. Because this will give information about the significance of the parameters. The significance of each parameter needs to be explained that 'a,' 'b,' 'ab,' etc. compared to which group.
  3. In the methods section, Line no. 83 "After day…" please verify is that after three days of treatment?
  4. Line No. 96, the write up is not correct. Please modify the sentence "blood was collected by cardiac puncture and mice tissues liver…….."
  5. Please mention the objectives used in Figure 2C and Figure 5C. Higher magnification of figure 5C may help to understand the results mentioned.
  6. Western blot analysis of PS-1 and PS-2 needs its corresponding β-actin. The same β-actin is used. If both are from the same membrane, then that needs to be mentioned in the figure legend.
  7. Typographical errors such as Figure legend 6, please delete 'E' in front of the legend.
  8. The figures' quality is not good, and this gives most of the information typed in the figures that are not clear to read (The axis numbers and title).

Author Response

Thank you for the valuable comments on this paper. We considered the comments carefully and the manuscript has been revised according to the comments.

Reviewer 1.

The authors Kim et al. demonstrated the effect of A. okamotoanum and its active compound Isoquercitrin (IQ) along with positive controls for anti-AD and anti-obesity. This article is part of their already published article Kim et al. 2019.

; Kim et al., (2019) reported that cognitive improvement effects of A. okamotoanum and isoquercitrin by regulation of oxidative stress in high fat diet (HFD)- and amyloid beta (Aβ)-induced mice model. In this study, we investigated protective effects of A. okamotoanum and isoquercitrin against obesity and amyloidogenesis in HFD- and Aβ-induced mice model. Therefore, this study has an originality and individuality.

  1. The rationale of using positive control mentioned in the discussion section. Please bring the write-up in the introduction section for a better understanding of readers. The write-up will provide information to readers why the donepezil and CLA used in the study.

; We added it in introduction (Page 2, Line 60-64)

[Introduction]

Donepezil and conjugated linoleic acid (CLA) are widely used in treatment of AD and obesity, respectively (Kniowles, 2006; Koba and Yanagita, 2014). Therefore, this study focused on the protective effects of EAO and isoquercitrin on the obesity and amyloidosis compared with donepezil and CLA, in HFD and Aβ-induced in vivo mice model.

[References]

Knowles, J. Donepezil in Alzheimer's disease: an evidence-based review of its impact on clinical and economic outcomes. Core Evid. 2006, 1, 195-219.

Koba, K.; Yanagita, T. Health benefits of conjugated linoleic acid (CLA). Obes. Res. Clin. Pract. 2014, 8, 525-532.

  1. The primary concern of the article is the number of mice/samples used in each parameter. Because this will give information about the significance of the parameters. The significance of each parameter needs to be explained that 'a,' 'b,' 'ab,' etc. compared to which group.

; We added number of mice (n = 6) in Materials & Methods (Page 2, Line 90; Page 3, Line 150) and Figure Legends. The results were analyzed using one-way ANOVA followed by Duncan’s multiple test (P < 0.05). Duncan’s multiple test is a multiple comparison among all groups in a statistical significance. We explained significance of each parameter in Materials & Methods and Figure Legends.

[Materials and Methods]

P < 0.05 was considered to indicate a statistically significant difference among all groups

[Figure Legends]

a~dMeans indicated with different letters are significantly different by Duncan’s multiple range test among all groups.

  1. In the methods section, Line no. 83 "After day…" please verify is that after three days of treatment?

; We revised it to ‘After consumption of HFD for 10 weeks’ in Materials and Methods section (Page 2, Line 89-90).

  1. Line No. 96, the write up is not correct. Please modify the sentence "blood was collected by cardiac puncture and mice tissues liver…….."

; We revised it in Materials and Methods (Page 3, Line 102-103).

[Materials and Methods]

The mice were overexposed to CO2 gas to be anaesthetized, and then blood was collected by cardiac puncture. Mice tissues such as liver, brain, and adipose tissue were perfused with saline solution and removed.

  1. Please mention the objectives used in Figure 2C and Figure 5C. Higher magnification of figure 5C may help to understand the results mentioned.

; We revised figure’s quality and added magnification in Materials and Methods (Page 3, Line 129).

  1. Western blot analysis of PS-1 and PS-2 needs its corresponding β-actin. The same β-actin is used. If both are from the same membrane, then that needs to be mentioned in the figure legend.

; Protein expressions such as PS-1 and PS-2 were from the same membrane. Therefore, we revised it in Figure 7.

  1. Typographical errors such as Figure legend 6, please delete 'E' in front of the legend.

; We revised it in Figure legend 6.

  1. The figures' quality is not good, and this gives most of the information typed in the figures that are not clear to read (The axis numbers and title).

; We revised figure’s quality. In addition, we added title of Figures.

Reviewer 2 Report

General comments:

The presented study investigated the protective effects of the natural product Acer okamotoanum and the containing bioactive flavonoid isoquercitrin on the development of obesity and Alzheimer’s disease. For investigations C57BL/6 wild type mice fed a high-fat diet were injected with amyloid beta representing a mouse model combining obesity and Alzheimer’s disease. The authors found protective effects on the development of obesity, obesity-induced fat liver and Alzheimer’s disease in the Acer okamotoanum and the isoquercitrin treated animals. As characteristic markers for obesity weight gain, adipocyte morphology, blood glucose, leptin and adiponectin were analysed, whereas steatosis and liver inflammation marker were determined to characterize the fat liver. The development of Alzheimer’s disease was measured by the expression of the β-and γ-secretase.

Major comments:

  • In your manuscript you showed protective effects on obesity, fat liver and Alzheimer’s disease. The relationship between obesity and fat liver is well known. Is there any literature around showing comorbidities of fat liver and Alzheimer’s disease? Are there signalling pathways shared by these diseases? Please include a related paragraph in your “discussion section”.
  • Please provide the number of mice used per group and for measurements in each figure or provide single data points for each mice in your graphs.
  • The overall quality of the graphs and pictures is poor. Consequently, the evaluation of your histology pictures is only possible to a limited extent. Please provide a better resolution of your pictures and graphs.
  • The SD in Figure 4 are large. Do you have an explanation for that?
  • Figure 5:
    • The inflammatory liver marker AST and ALT are given in “karmen”. Please provide the more common unit “U/L”
    • In Figure 5 you investigated the liver pathology. How did you characterize and quantify the steatosis level of your sections? Did you quantify ballooning, fibrosis, and accumulation of lipids or inflammatory marker in the liver sections to verify the high fat diet-induced fat liver?
  • Figure 6:
    • In your analysis BACE-1 protein expression is decreased in your treatment groups. However, the BACE-1 blot does not show that. Please show representative Western blots for BACE-1.
    • Which part of the brain has been used for the Western blot analysis
  • Figure 7:
    • Please show representative Western blots for PS-1/2.

Minor comments:

  • Line 90: “after day, mice were divided…”. Please provide the day.
  • Line 316: anti-obesity and anti-diabetic instead of antiobesity and antidiabetic.
  • Line 364/65, 369, 377/78: Please rewrite the sentences, considering the grammar.
  • Method section: Please provide “x g” instead of “RPM” for centrifugation.
  • Please provide uncropped Western blots.

Author Response

Thank you for the valuable comments on this paper. We considered the comments carefully and the manuscript has been revised according to the comments.

Reviewer 2

Major comments:

  1. In your manuscript you showed protective effects on obesity, fat liver and Alzheimer’s disease. The relationship between obesity and fat liver is well known. Is there any literature around showing comorbidities of fat liver and Alzheimer’s disease? Are there signalling pathways shared by these diseases? Please include a related paragraph in your “discussion section”.

; Thank you for the valuable comments. We added it in discussion section (Page 11, Line 331-340).

[Discussion]

Nonalcoholic fatty liver disease (NAFLD) is characterized by accumulation of fat in liver, and it is known as associated with AD. NAFLD mice induced by HFD showed cognitive and memory impairment in behavior tests (Pinçon et al., 2019). The HFD-fed NAFLD mice showed decrease of Aβ clearance in the liver, and then it induced aggravation of cognitive impairment by accumulation of Aβ plaques in the brain (Pinçon et al., 2019; Estrada et al., 2019). The HFD-fed mice significantly increased circulating Aβ level by decreasing low-density lipoprotein receptor-1 (Estrada et al., 2019; Kim et al., 2016). In addition, fatty liver increased neuroinflammation by increasing microgliosis, toll-like receptors, and inflammatory cytokines in the brain (Kim et al., 2016). Furthermore, fatty liver-induced mice increased neuronal oxidative stress by increasing gene expression of NOX2 (Pinçon et al., 2019). Therefore, fatty liver is closely related to AD progression by regulation of Aβ clearance, neuroinflammation, and oxidative stress.

[References]

Pinçon, A.; Montgolfier, O.D.; Akkoyunlu, N.; Daneault, C.; Pouliot, P.; Villeneuve, L.; Lesage, F.; Levy, B.I.; Thorin-Trescases, N.; Thorin, É.; Matthieu, Ruiz, M. Non-alcoholic fatty liver disease, and the underlying altered fatty acid metabolism, reveals brain hypoperfusion and contributes to the cognitive decline in APP/PS1 mice. Metabolites. 2019, 9, 104.

Estrada, L.D.; Ahumada, P.; Cabrera, D.; Arab, J.P. Liver dysfunction as a novel player in Alzheimer's progression: looking outside the brain. Front Aging Neurosci. 2019, 11, 174.

Kim, D.G.; Krenz, A.; Toussaint, L.E.; Maurer, K.J.; Robinson, S.A.; Yan, A.; Torres, L.; Bynoe, M.S. Non-alcoholic fatty liver disease induces signs of Alzheimer’s disease (AD) in wild-type mice and accelerates pathological signs of AD in an AD model. J Neuroinflammation. 2016, 13, 1.

  1. Please provide the number of mice used per group and for measurements in each figure or provide single data points for each mice in your graphs.

; We added number of mice (n = 6) in Materials & Methods (Page 2, Line 90; Page 3, Line 150) and Figure Legends.

  1. The overall quality of the graphs and pictures is poor. Consequently, the evaluation of your histology pictures is only possible to a limited extent. Please provide a better resolution of your pictures and graphs.

; We revised figure’s quality and added magnification in Materials and Methods (Page 3, Line 129).

  1. The SD in Figure 4 are large. Do you have an explanation for that?

; The levels of leptin and adiponectin in Figure 4 were represented mean value of 6 mice in each group. This standard deviation is relatively large, compared with other parameters. Other studies supported that the levels of leptin and adiponectin varies between mice’s individual compared to other parameters (Ohno et al., 2015; Liang et al., 2014).

[References]

Ohno T, Shimizu M, Shirakami Y, Baba A, Kochi T, Kubota M, Tsurumi H, Tanaka T, Moriwaki H. Metformin suppresses diethylnitrosamine-induced liver tumorigenesis in obese and diabetic C57BL/KsJ-+Leprdb/+Leprdb mice. PLoS One. 2015 Apr 16;10(4):e0124081.

Liang X, Pei H, Ma L, Ran Y, Chen J, Wang G, Chen L. Synthesis and Biological Evaluation of Novel Urea- and Guanidine-Based Derivatives for the Treatment of Obesity-Related Hepatic Steatosis. Molecules. 2014 May 15;19(5):6163-83.

  1. Figure 5: The inflammatory liver marker AST and ALT are given in “karmen”. Please provide the more common unit “U/L”

; We revised unit from karmen to IU/L in Figure 5.

  1. In Figure 5 you investigated the liver pathology. How did you characterize and quantify the steatosis level of your sections? Did you quantify ballooning, fibrosis, and accumulation of lipids or inflammatory marker in the liver sections to verify the high fat diet-induced fat liver?

; Thank you for the valuable comments. We added it in Discussion (Page 10-11, Line 318-325).

[Discussion]

To investigate fatty accumulation and liver toxicity of samples, we investigated the liver pathology by H&E staining. This result showed approximate fatty liver lesions and toxicity. As shown in Figure 5C, the HFD-fed control group showed an accumulation of numerous fatty droplets and hepatic steatosis, compared with ND-fed normal group. The administration of sample groups attenuated accumulation of liver fatty droplets and hepatic steatosis. Further study is needed to hepatoprotective effects and its molecular mechanisms of samples by quantify ballooning, fibrosis, and accumulation of lipids or measurement of inflammatory markers.

  1. Figure 6: In your analysis BACE-1 protein expression is decreased in your treatment groups. However, the BACE-1 blot does not show that. Please show representative Western blots for BACE-1.

; We measured and quantified BACE-1/beta-actin from western band. Treatment of ethyl acetate fraction from Acer okamotoanum, isoquecitrin, donepezil, and CLA significantly inhibited BACE-1 protein expression, compared with control group.

  1. Which part of the brain has been used for the Western blot analysis

; We provided part of the brain in Materials and Methods (Page 3, Line 132).

[Materials and Methods]

Whole brain tissue was homogenized and lysed in ice-cold RIPA lysis buffer ~

  1. Figure 7: Please show representative Western blots for PS-1/2.

; We added PS-1/2 ratio in Figure 7.

Minor comments:

  1. Line 90: “after day, mice were divided…”. Please provide the day.

; We revised it to ‘After consumption of HFD for 10 weeks’ in Materials and Methods section (Page 2, Line 89-90).

  1. Line 316: anti-obesity and anti-diabetic instead of antiobesity and antidiabetic.

; We revised it in all manuscript.

  1. Line 364/65, 369, 377/78: Please rewrite the sentences, considering the grammar.

; We rewrite the sentences and use a professional English editing service.

  1. Method section: Please provide “x g” instead of “RPM” for centrifugation.

; We provide xg for centrifugation in Materials and Method (Page 3, Line 119, 133).

  1. Please provide uncropped Western blots.

; We provided uncropped Western blots bands.

Round 2

Reviewer 1 Report

The authors carried out the concern raised. However, few minor corrections required.

  1. The significance of each parameter needs to be explained that 'a,' 'b,' 'ab,' etc. compared to which group. For example, Figure 2D 'b' may be significant with the Normal and control group. However, the same 'b' is not significant with IQ, CLA groups. To differentiate this, the authors need to explain in the figure legend -‘b' significant with the Normal and control group. This will give the readers a clear understanding of 'b' significant against which group. Likewise, other letters also need to explain.
  2. The authors mentioned in reply to the comments, the same membrane was used for PS-1 and PS-2 blot. Please provide this information in the figure legend.
  3. The font size needs to be bigger for all the information typed in all the graphs. It is very small and heard to read.

Author Response

Reviewer 1.

The authors carried out the concern raised. However, few minor corrections required.

The significance of each parameter needs to be explained that 'a,' 'b,' 'ab,' etc. compared to which group. For example, Figure 2D 'b' may be significant with the Normal and control group. However, the same 'b' is not significant with IQ, CLA groups. To differentiate this, the authors need to explain in the figure legend -‘b' significant with

the Normal and control group. This will give the readers a clear understanding of 'b' significant against which group. Likewise, other letters also need to explain.

; We explained significance compared to normal and control group, respectively. We revised it in Figure and Figure legend.

[Figure legend]

*P < 0.05 vs. normal; #P < 0.05 vs. control.

The authors mentioned in reply to the comments, the same membrane was used for PS-1 and PS-2 blot. Please provide this information in the figure legend.

; We added it in Figure 7 legend.

[Figure 7 legend]

Same membrane was used for expressions of PS-1 and PS-2 with stripping and re-probing.

The font size needs to be bigger for all the information typed in all the graphs. It is very small and heard to read.

; We revised the font size of all the graphs to bigger.

Reviewer 2 Report

The authors improved the manuscript by adapting most of the comments. However, the authors claim that uncropped Western blots are presented in the manuscript. As far as I see, no changes have been made to the original manuscript. Presenting uncropped blots means showing the complete developed image with the size of the bands according to the marker, not just the bands of interest to show the quality of the antibody/your Western blot. Therefore, please provide uncropped Western blots.

Author Response

Reviewer 2

The authors improved the manuscript by adapting most of the comments. However, the authors claim that uncropped Western blots are presented in the manuscript. As far as I see, no changes have been made to the original manuscript. Presenting uncropped blots means showing the complete developed image with the size of the bands according to the marker, not just the bands of interest to show the quality of the antibody/your Western blot. Therefore, please provide uncropped Western blots.

; As shown in Fig. 1, we investigated protein expressions of eight groups by western blotting. Eight groups as follows: N, normal diet + 0.9% NaCl injection + oral administration of water; NA, normal diet + Aβ25-35 injection + oral administration of water; H, high fat diet (HFD) + 0.9% NaCl injection + oral administration of water; C, HFD + Aβ25-35 injection + oral administration of water; AO, HFD + Aβ25-35 injection + oral administration of EtOAc fraction from Acer okamotoanum; IQ, HFD + Aβ25-35 injection + oral administration of isoquercitrin; DO, HFD + Aβ25-35 injection + oral administration of donepezil; CLA, HFD + Aβ25-35 injection + oral administration of CLA.

Figure 1. Western blot band

However, to explain this study, we used protein expressions of six groups from Fig. 1. Six groups as follows: N, C, AO, IQ, DO, CLA. Therefore, we used cropped Western blots band in this study.

This manuscript is a resubmission of an earlier submission. The following is a list of the peer review reports and author responses from that submission.